# The CfSnt2-Dependent Deacetylation of Histone H3 Mediates Autophagy and Pathogenicity of *Colletotrichum fructicola*

**DOI:** 10.3390/jof8090974

**Published:** 2022-09-18

**Authors:** Yuan Guo, Zhenhong Chen, He Li, Shengpei Zhang

**Affiliations:** 1College of Forestry, Central South University of Forestry and Technology, Changsha 410004, China; 2Key Laboratory of National Forestry and Grassland Administration on Control of Artificial Forest Diseases and Pests in South China, Central South University of Forestry and Technology, Changsha 410004, China; 3Hunan Provincial Key Laboratory for Control of Forest Diseases and Pests, Central South University of Forestry and Technology, Changsha 410004, China; 4Key Laboratory for Non-wood Forest Cultivation and Conservation of Ministry of Education, Central South University of Forestry and Technology, Changsha 410004, China

**Keywords:** histone deacetylation, autophagy, pathogenicity, *C. fructicola*

## Abstract

*Camellia oleifera* is one of the most valuable woody edible-oil crops, and anthracnose seriously afflicts its yield and quality. We recently showed that the CfSnt2 regulates the pathogenicity of *Colletotrichum fructicola*, the dominant causal agent of anthracnose on *C. oleifera*. However, the molecular mechanisms of CfSnt2-mediated pathogenesis remain largely unknown. Here, we found that CfSnt2 is localized to the nucleus to regulate the deacetylation of histone H3. The further transcriptomic analysis revealed that CfSnt2 mediates the expression of global genes, including most autophagy-related genes. Furthermore, we provided evidence showing that CfSnt2 negatively regulates autophagy and is involved in the responses to host-derived ROS and ER stresses. These combined functions contribute to the pivotal roles of CfSnt2 on pathogenicity. Taken together, our studies not only illustrate how CfSnt2 functions in the nucleus, but also link its roles on the autophagy and responses to host-derived stresses with pathogenicity in *C. fructicola*.

## 1. Introduction

*Camellia oleifera* is one of the most valuable woody edible-oil crop and has been grown in China for over 2300 years [1]. The oil extracted from its seed has been widely utilized in cooking, lubricant, and cosmetics [2,3]. Through the planting areas of *C*. *oleifera* was 4.39 million hectares in China, and anthracnose afflicted its high yield and quality [4,5,6]. We recently showed that the dominant causal agent of anthracnose on *C. oleifera* is *Colletotrichum fructicola*, and CfSnt2 regulates the pathogenicity of *C. fructicola* through mediating the formation of functional appressorium [6,7]. However, the molecular mechanisms of CfSnt2-mediated pathogenesis is unclear.

The Snt2 protein was initially identified and named for the DNA binding SANT domain, and it was widely conserved in fungal species [8,9]. The ScSnt2 acts as an E3 ubiquitin ligase to regulate the degradation of excess histone proteins H3 and H4 [8]. The yeast ScSnt2 also recruits histone deacetylase ScRpd3 to the promoters of genes related to oxidative stress for the transcriptional response [10]. In phytopathogenic fungi, there are only limited studies on Snt2 proteins. In *Fusarium oxysporum* and *Neurospora crassa*, Snt2 proteins mediate fungal respiration and reactive oxidative stress by regulating the expression of oxidase genes [11]. Further study in *F. oxysporum* firstly revealed the roles of Snt2 in fungal autophagy and pathogenicity [12]. Furthermore, in *Fusarium graminearum*, Snt2 links histone deacetylation, autophagy, and pathogenicity [13].

Macroautophagy/autophagy is a highly conserved process, by which proteins, membranes, and organelles are degraded in the vacuole (lysosome) and recycled to satisfy energy needs [14,15,16,17]. A growing body of evidence points to the pivotal roles of autophagy in fungal pathogenicity [18,19,20]. In *Magnaporthe oryzae*, autophagy is important for the formation of functional appressoria, lipid metabolism, and pathogenicity [21,22]. In *F. graminearum*, targeted gene deletion of autophagy-related genes *FgATG9* showed decreased pathogenicity in wheat, which was caused by the severe defects in autophagy and lipid metabolism [23]. We also previously revealed that the *CfATG8* gene deletion mutant in *C. fructicola* was non-pathogenic to *C*. *oleifera* [24].

Recent studies in the phytopathogenic fungus have shown the critical regulatory roles of acetylation levels for autophagy-dependent pathogenicity. The histone acetyltransferase MoHat1 governs autophagy-dependent pathogenicity through acetylates MoAtg3 and MoAtg9 in *M. oryzae* [25]. The *M. oryzae* histone acetyltransferase MoGcn5 links the acetylation of MoAtg7 between the light-induced autophagy and pathogenicity [26]. The *F. graminearum* FgGcn5 acetylates histone H3 and FgAtg8, governing the function associated with autophagy and pathogenicity [27]. Our recent study also revealed that CfGcn5 regulates development and pathogenicity in *C. fructicola* [28]. Whether acetylation modifications are related with autophagy-dependent pathogenicity in *C. fructicola* is unknown.

In the present study, we revealed that CfSnt2 is localized to nucleus for the deacetylation of histone H3, resulting in its regulation of global genes expression. Meanwhile, we revealed that CfSnt2 mediates the expression of multiple *CfATG* genes and negatively regulates autophagy. In addition, we also demonstrated that CfSnt2 is involved in the tolerance of oxidative and endoplasmic reticulum (ER) stress.

## 2. Results

### 2.1. CfSnt2 Is Localized to the Nucleus and Mediates H3 Deacetylation

To further dissect the function of CfSnt2, we fused a GFP tag to the C-terminus of CfSnt2 and found that CfSnt2-GFP showed a slight fluorescence affecting visualization. Therefore, we fused the GFP tag to the N-terminus of CfSnt2 and found that GFP-CfSnt2 was clearly visible as spot green fluorescence. To investigate whether the spot green fluorescence represents the nucleus, we fused the RFP tag to histone H1, a widely used nucleus marker. The co-localization of GFP-CfSnt2 and H1-RFP showed that CfSnt2 localized to the nucleus in the mid and tip regions of hyphae and conidia (Figure 1A,B). We then wondered whether CfSnt2 mediates H3 deacetylation in the nucleus. The previous study in *M. oryzae* revealed that H3K18 acetylation (H3K18ac) mediated by the histone acetyltransferase MoTig1 is essential for pathogenicity [29]. The immunoblotting assays, using the anti-H3K18ac and anti-H3 antibodies, revealed the increased acetylation levels of histone H3K18 in Δ*Cfsnt2* mutant, compared with the wide type (WT) and complemented strains Δ*Cfsnt2/CfSNT2-1* and Δ*Cfsnt2/CfSNT2-2* (Figure 1C). Since the complemented strains Δ*Cfsnt2/CfSNT2-1* and Δ*Cfsnt2/CfSNT2-2* showed similar phenotypes, we selected Δ*Cfsnt2/CfSNT2-1* for our further study.

### 2.2. Transcriptomic Analysis of the WT and ΔCfsnt2 Mutant

To test the roles of CfSnt2-mediated H3 deacetylation in the nucleus, we carried out transcriptome analysis for the WT and Δ*CfSnt2* mutant by RNA sequencing (RNA-seq). Three biological replicates were established for each strain, and six RNA-seq data sets were generated, among which, more than 90% reads were mapped to the genome of *C. fructicola* (GenBank accession number: GCA_000319635.2). All raw data were submitted to the NCBI SRA database, with accession numbers SRR19052615, SRR19052614, and SRR19052613 for WT 1-3 and SRR19052618, SRR19052617, and SRR19052616 for Δ*Cfsnt2* 1-3. Principal component analysis (PCA) of WT and Δ*Cfsnt2* mutant revealed a clear separation of the two tested strains, as well as the proximity of the biological replicates (Figure 2A). Gene expression analysis revealed that a total of 9572 predicted genes were expressed in both the WT and Δ*Cfsnt2* mutant, and 990 and 924 predicted genes were specifically expressed in the WT and Δ*Cfsnt2* mutant (Figure 2B). Differentially expressed genes (DEGs) analysis showed that 2408 predicted genes were up-regulated and 2565 predicted genes were down-regulated at least two-fold (*p* < 0.01) in the Δ*Cfsnt2* mutant (Figure 2C).

### 2.3. Gene Ontology Enrichment of the DEGs

All the DEGs were further analyzed by gene ontology (GO) enrichment (*p* < 0.01), and a total of 25 GO enrichment terms were categorized into three main categories: biological process, cellular component, and molecular function (Figure 3). The biological process contains nine terms, including translation, response to oxidative stress, amine metabolic process, cellular amide metabolic process, fatty acid biosynthetic process, replication fork protection, amide biosynthetic process, invasive growth in response to glucose, and limitation peptide metabolic process. The cellular component contains three terms, including ribosome, integral component of membrane, and small ribosomal subunit. The molecular function contains 13 terms, including structural constituent of ribosome, N-acetyltransferase activity, ATPase activity, coupled to transmembrane movement of substances, pyridoxal phosphate binding, aspartic-type endopeptidase activity, molybdenum ion binding, glutamate-ammonia ligase activity, glutaminase activity, ATPase activity, alpha-1,6-mannosyltransferase activity, unfolded protein binding, copper ion binding, and nucleotidyltransferase activity.

### 2.4. KEGG Pathway Enrichment of the DEGs

Significantly enriched (*p* < 0.01) KEGG pathways of the DEGs were also analyzed. The results showed that 10 KEGG pathways were significantly enriched, including one carbon pool by folate, arginine biosynthesis, nitrogen metabolism, steroid biosynthesis, ribosome, nicotinate and nicotinamide metabolism, glyoxylate and dicarboxylate metabolism, alanine, aspartate and glutamate metabolism, butanoate metabolism, and glycine, serine, and threonine metabolism (Figure 4).

### 2.5. CfSnt2 Mediates the Expression of Multiple CfATG Genes

To verify the gene expression profiles in the transcriptomic data, three up-regulated and three down-regulated genes were also analyzed by qRT-PCR. Though the magnitude of fold changes showed a slight change to the transcriptomic data, the qRT-PCR data were all consistent with that from the transcriptomic data (Figure 5A). In the RNA-seq data, we also found most autophagy-related genes (Appendix A), particularly for the homologs of pathogenicity-associated macro-autophagy in *M. oryzae* [21], were among the DEGs. The qRT-PCR analysis demonstrated the upregulation of 13 *CfATGs* among 15 genes, including *CfATG1*, *CfATG2*, *CfATG3*, *CfATG4*, *CfATG6*, *CfATG7*, *CfATG8*, *CfATG9*, *CfATG10*, *CfATG12*, *CfATG13*, *CfATG15*, and *CfATG18* (Figure 5B). These results revealed that CfSnt2 mediates the expression of most autophagy-related genes, and we hypothesized its roles in autophagy.

### 2.6. CfSnt2 Is Involved in the Rapamycin Response

To demonstrate our hypothesis, we firstly test whether CfSnt2 is involved in the response of rapamycin, which induces autophagy via the Tor (target of rapamycin) signaling pathway [30,31]. When exposed to 25 nM rapamycin, the Δ*Cfsnt2* mutant exhibited a significantly higher inhibition rate than that of WT and Δ*Cfsnt2/CfSNT2* (Figure 6A,B). The results indicated that CfSnt2 is involved in the rapamycin response.

### 2.7. CfSnt2 Negatively Regulates Autophagy

To examine how CfSnt2 mediates autophagy, the fusion gene *GFP-CfATG8*, homologs of which were widely used as markers for macro-autophagy in other organisms [32,33], were introduced into the WT and Δ*Cfsnt2*. When grown in PDA-rich medium, Δ*Cfsnt2* mutant proved to be more autophagosomes than WT (Figure 7A,B). We further assessed autophagic flux by immunoblot in MM-N condition, another way to induce autophagy [34]. The free GFP (26 kDa) and full length GFP-CfAtg8 (46 kDa) could easily be detected in both the WT- and Δ*Cfsnt2* mutant-expressing *GFP-CfATG8* strains. Before MM-N induction, WT showed lower amounts of free GFP than GFP-CfAtg8, but the Δ*Cfsnt2* mutant showed comparable free GFP to GFP-CfAtg8. Upon MM-N induction for 4 h, Δ*Cfsnt2* mutant contained a slight GFP-CfAtg8, which contrasted with the still large amounts of GFP-CfAtg8 in WT (Figure 7C). The level of autophagy was also evaluated by measuring free GFP, compared with the total of free GFP and intact GFP-CfAtg8 together. The ratio of GFP in Δ*Cfsnt2* mutant was significantly higher than that in WT, both in the non-induction and induction conditions (Figure 7D), indicating the increased autophagy in Δ*Cfsnt2* mutant. The results revealed that CfSnt2 negatively regulates autophagy.

### 2.8. CfSnt2 Regulates Pathogenicity and the Responses to Oxidative and ER Stresses

CfSnt2 is involved in pathogenicity in *C. fructicola* (Figure 8A,B), partially due to its roles in the formation of functional appressoria [7]. To explore other underlying mechanism in the pathogenesis, we found that the response to oxidative stress and unfolded protein binding terms are among the GO enrichment of the DEGs. The reactive oxygen species (ROS) is a general defense response for the plants in plant–pathogen interactions, and it is necessary for the successful infection of pathogens to overcome host-derived ROS [35,36]. Additionally, the previous studies showed that pathogens must also activate the unfolded protein response to face host-derived ER stress during infection [37,38]. Thus, we used H_2_O_2_ and dithiothreitol (DTT) to mimic the host-derived ROS and ER stress, respectively. We found that Δ*Cfsnt2* mutant were more sensitive to both H_2_O_2_ and DTT stresses, compared with WT and Δ*Cfsnt2/CfSNT2* (Figure 8C,D). These results indicated that CfSnt2 mediates pathogenicity and the responses to oxidative and ER stresses.

## 3. Discussion

*Colletotrichum* spp. is ranked as the eighth most important fungal phytopathogens in plant pathology and causes anthracnose disease on almost every crop [39]. Despite the economical and genetic importance, its pathogenesis remains in a state that is largely unclear, particularly for the species of *C. fructicola*. We previously revealed that CfSnt2 regulates the pathogenicity by mediating the formation of functional appressorium in *C. fructicola*, but the underlying molecular mechanisms are unknown [3]. In this study, we not only provided evidence regarding how the CfSnt2 mechanism regulates global gene expressions, but also highlighted its roles in the pathogenicity-related autophagy and stress responses.

The yeast ScSnt2 protein was localized to the promoters of the stress response genes in the nucleus [10]. The *M. oryzae* MoSnt2 protein was also localized to the nucleus in different developmental stages [13]. Consistent with these studies, our research revealed that CfSnt2 was localized to the nucleus in the hyphae and conidia of *C. fructicola*, thus indicating its conserved regulatory roles in the nucleus. The further decreased acetylation level of H3K18 supports its functions on histone deacetylation in nucleus.

Acetylation and deacetylation of histones are pivotal epigenetic modifications for the transcription of genes [40,41]. Over the past decade, growing cases have indicated that the epigenetic acetylation and deacetylation of histones share important functions in fungal pathogenesis by regulating gene expression and other mechanisms [42,43]. The large amount of DEGs (2408 up-regulated and 2565 down-regulated) in the Δ*Cfsnt2* mutant of the transcriptomic data revealed its roles in gene expression through histone deacetylation. However, histone deacetylation is generally associated with transcriptional inactivation [40,44]. The slightly more down-regulated genes, rather than up-regulated genes, indicated that Cfsnt2 might also mediate the acetylation of other histones. This was apparent in the recent study of *F. graminearum*, which demonstrated that FgFng3 not only regulates H3 acetylation, but also mediates H4 deacetylation [45]. This might be interesting for illustrating the link between CfSnt2-mediated acetylation and gene expression, and further studies are warranted.

The GO and KEGG enrichment analysis of the DEGs indicated that CfSnt2 regulates the genes that are mainly involved in translation, ribosomal biogenesis, amino acid, and carbohydrate biosynthesis/metabolism, thus revealing the basic roles of CfSnt2 in the physiological process of *C. fructicola*. Particularly, most of the pathogenicity-related *CfATG* genes were significantly increased in the Δ*Cfsnt2* mutant, thus forecasting its roles in autophagy. The contributions of CfSnt2 on rapamycin stress response further confirmed this hypothesis. Indeed, the autophagosome abundance was increased in the Δ*Cfsnt2* mutant, which showed similar phenotype to the *F. graminearum* and *F.oxysporum* Δsnt2 mutant [12,13]. Again, the furthermore immunoblot analysis also demonstrated the negative role of CfSnt2 in autophagy. Multiple evidences point to the essential roles of autophagy in fungal pathogenicity [21,38]. The disruption of autophagic homeostasis in the Δ*Cfsnt2* mutant might be one reason for the pathogenicity defects. Additionally, the Δ*Cfsnt2* mutant is sensitive to the mimicked host-derived ROS and ER stresses, which might be another reason for its pathogenicity defects.

In summary, we demonstrated that CfSnt2 is localized to the nucleus, whereby it reduces the acetylation abundance of histone H3, resulting in the changed expression of multiple genes associated with developmental processes, including pathogenicity-related autophagy and ROS and ER stress responses. The identification of epigenetic histone and non-histone modifications for CfSnt2 in pathogenesis remains interesting work, and further studies are highly warranted.

## 4. Materials and Methods

### 4.1. Strains and Culture Conditions

The *C. fructicola* CFLH16 strain was used as WT in this study. The *CfSNT2* gene deletion mutant was generated by the standard one-step gene replacement strategy in our previous study [7]. Briefly, two~1.0 kb sequences flanking the *CfSNT2* gene were PCR amplified and overlapped to the flanks of hygromycin resistance cassette (1.4 kb), respectively. After sequencing, the~3.4 kb fragment was transformed into the protoplasts of WT for gene deletion. The complement fragment, which contains the entire *CfSNT2* gene and its native promoter region, was amplified by PCR and inserted into pYF11 (bleomycin resistance) to complement the Δ*Cfsnt2* mutant. All strains were cultured on potato dextrose agar (PDA) agar plates in the darkness at 28 °C. Liquid potato dextrose broth (PDB) medium was used to culture the mycelia for DNA, RNA, and protein extraction.

To test the autophagic flux, the strains were incubated in MM-N medium (0.52 g KCl, 0.152 g MgSO_4_·7H_2_O, 1.52 g KH_2_PO_4_, 0.01 g vitamin B1, 1 mL trace elements, and 10 g D-glucose in 1 L of distilled water).

### 4.2. Generation of the CfSnt2-GFP, H1-RFP, and GFP-CfSnt2 Constructs

For generating CfSnt2-GFP, PCR fragment of the~1.5 kb CfSNT2 native promoter and gene coding sequence was co-transformed with XhoI-digested pYF11 into yeast strain XK1-25 [46]. Then, the fused CfSnt2-GFP construct was transformed into the Escherichia coli Trelief^TM^ 5α (TsingKe Biological Technology Co., Ltd, Beijing, China) and further sequenced. Similarly, the H1-RFP construct was generated by cloning the H1 gene into the pHZ126 vector (hygromycin resistance). For generating GFP-CfSnt2, the native promoter, 0.7 kb GFP fragment, and full-length of CfSNT2 were successively fused together and then inserted into the pYF11.

### 4.3. Localization Assays

The hyphae and conidia of the strain co-transformed with GFP-CfSnt2, and H1-RFP were visualized under a fluorescent microscope (ZEISS, Axio Observer. A1, Jena, Germany).

### 4.4. Protein Extraction and Immunoblot Analysis

The protein was extracted as described previously [38]. Briefly, the mycelia were filtered with Mirocloth, blotted dry, and ground into powder in liquid nitrogen using a mortar and pestle. The mycelial powder was then re-suspended in 1 mL RIPA lysis buffer (1%Triton X-100, 1% sodium deoxycholate, 0.1% SDS, (EpiZyme, PC101)) with 10 uL protease inhibitor cocktail (EpiZyme, GRF101), followed by being shaken once each (10 min) for 30 min on ice. The samples were centrifuged under 12,000× *g* at 4 °C for 20 min, and the supernatant liquids were acquired as total proteins. The immunoblot analysis was performed with 10% SDS-PAGE, followed by GFP (1:10,000, ABways, AB0045), α-H3K18ac (1:3000, Beyotime, AF5617), and α-H3 (1:2000, Cell Signaling Technology, 4499) antibodies.

### 4.5. Transcriptome Sequencing and Analysis

The RNA of the collected sample was extracted from WT and Δ*Cfsnt2* mutant with three biological replicates for each strain. Transcriptome sequencing was carried out by the Biomarker Technologies Company (Beijing, China). The libraries were sequenced on an Illumina HiSeq platform with 150 bp paired-end reads, followed by the alignment to the reference genome by Hisat2 [47]. DEG analysis were performed with an adjusted *p* < 0.01 and fold change ≥2 by DESeq2 [48].

### 4.6. QRT-PCR Assays

Total RNA were reverse transcribed into first-strand cDNA. The qRT-PCR using 2×TSINGKE^®^ Master qPCR Mix (SYBR Green I, TsingKe Biological Technology Co., Ltd, Beijing, China) with primers (Appendix A) was run on the ABI QuantStudio 3. The *ACTIN* gene was used as an internal reference for relative expression analysis. The experiment was repeated in at least 3 biological experiments independently, with 3 replicates.

### 4.7. Stress Response Analysis

The strains were cultured on PDA, and PDA added with 25 nM rapamycin, 10 mM or 20 mM H_2_O_2_, and 5 mM DTT. After 3 days of growth, the colony diameters were examined, and the inhibited rates were analyzed statistically.

### 4.8. Pathogenicity Assays

The pathogenicity assays were performed as previously described [49]. The mycelial plugs of the strains were inoculated onto the margin area of the wounded *C. oleifera* leaves. After incubation in a humidity plate with 12 h light and 12 h dark cycles for 4 days, the inoculated leaves were photographed and analyzed by Image J.

### 4.9. Statistical Analysis

Each result was presented as the mean ± SD of three replicates. The significance of differences between samples were analyzed by ANOVA (analysis of variance) with Duncan’s new multiple range test. The level of significance was set at *p* < 0.01 or *p* < 0.05.

## Figures and Tables

**Figure 1 jof-08-00974-f001:**
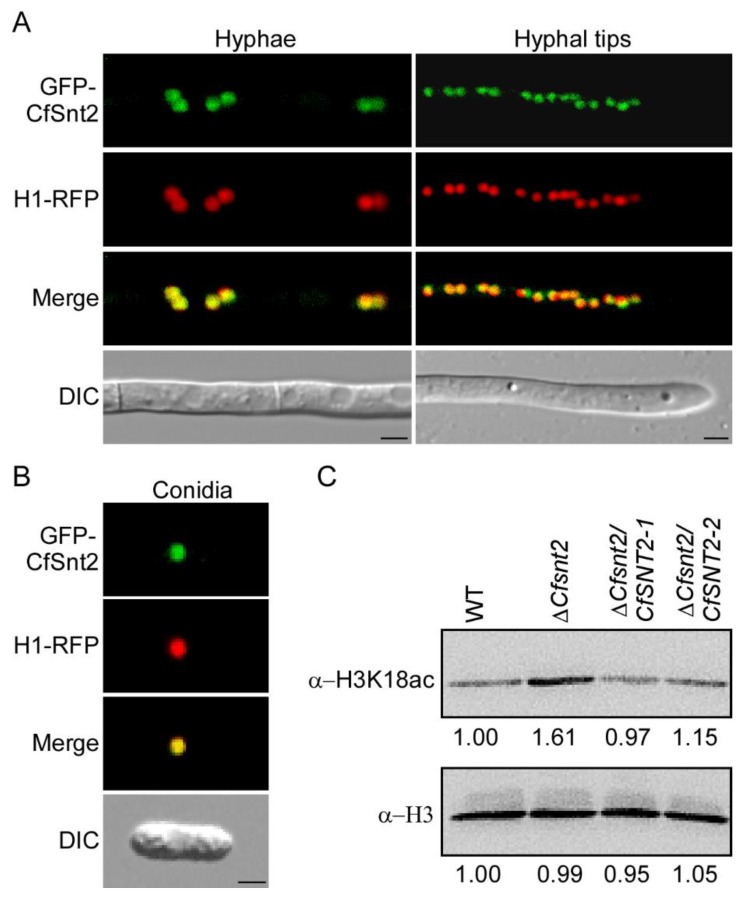
CfSnt2 is localized to the nucleus and mediates the deacetylation of H3K18. (**A**) The localization pattern of CfSnt2 in the mid and tip regions of the hyphae. The merged images of GFP-CfSnt2 and H1-RFP indicated that GFP-CfSnt2 is localized to the nucleus. (**B**) Localization pattern of CfSnt2 in conidia. Bar = 5 μm. (**C**) Total proteins extracted from mycelia of WT, ΔCfsnt2 mutant and complemented strains Δ*Cfsnt2/CfSNT2-1* and Δ*Cfsnt2/CfSNT2-2* were immunoblotted with the anti-H3K18ac and anti-H3 antibodies. The intensity of the protein bands were analyzed by Image J, and the intensity of WT was defined as a reference, with 1.00.

**Figure 2 jof-08-00974-f002:**
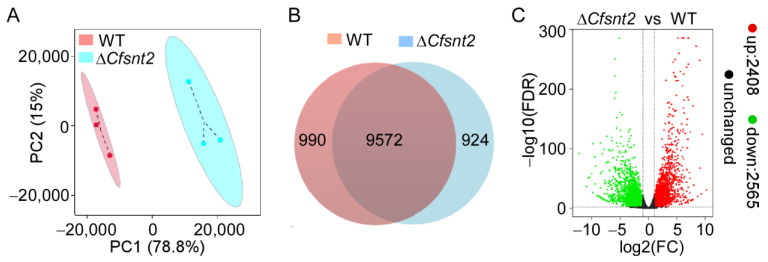
Transcriptomic analysis of the WT and Δ*Cfsnt2* mutant. (**A**) Principal component analysis (PCA) of WT and Δ*Cfsnt2* mutant. (**B**) Global view of expressed genes in WT and Δ*Cfsnt2* mutant. (**C**) Volcano plot of DEGs between WT and Δ*Cfsnt2* mutant. Red dots indicate upregulated genes; green dots indicate downregulated genes; black dots indicate unchanged genes.

**Figure 3 jof-08-00974-f003:**
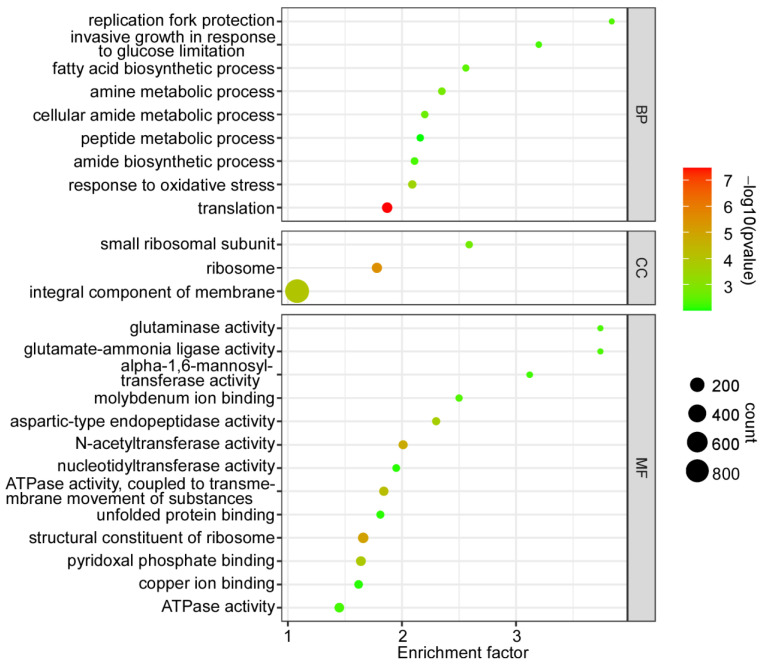
Gene ontology enrichment of the DEGs. The significant enrichment (*p* < 0.01) of the gene ontology (Go) terms are displayed in the diagram. The sizes of the dots indicate numbers, and the colors of the dots indicate *p*-value. BP: biological process; CC: cellular component; MF: molecular function.

**Figure 4 jof-08-00974-f004:**
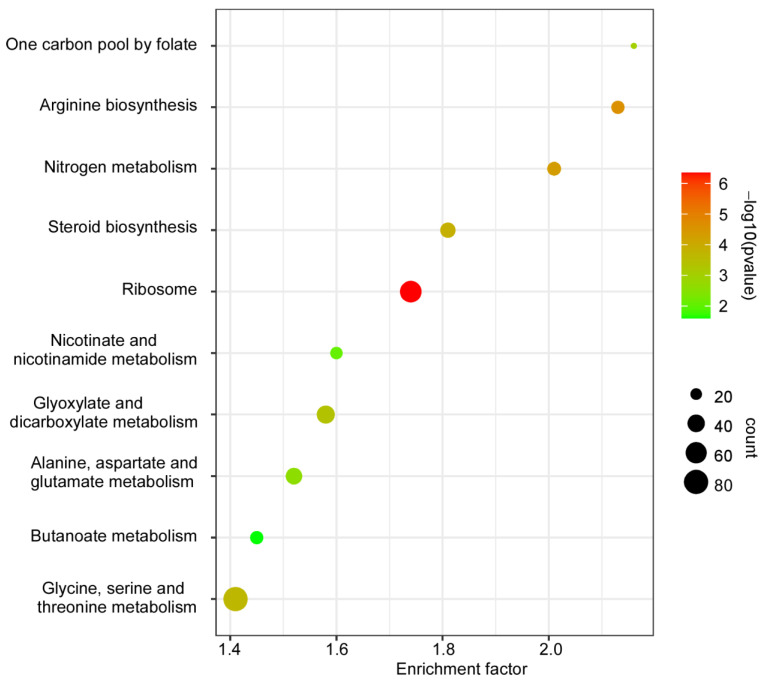
KEGG pathway enrichment of the DEGs. The significant enrichment (*p* < 0.01) of the KEGG pathways are displayed in the diagram. The sizes of the dots indicate numbers, and the colors of the dots indicate *p*-value.

**Figure 5 jof-08-00974-f005:**
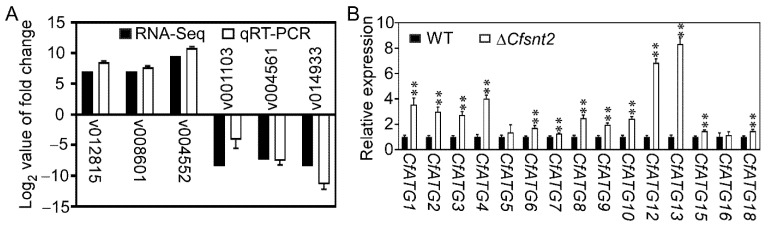
The gene expression levels in WT and Δ*Cfsnt2* mutant. (**A**) QRT-PCR analysis of the expression levels for selected 3 up-regulated genes and 3 down-regulated genes in the transcriptomic data of WT and Δ*Cfsnt2* mutant. (**B**) QRT-PCR analysis of the expression levels of *CfATG* genes in WT and Δ*Cfsnt2* mutant. Error bars indicate standard deviation (SD) of three biological replicates (** *p* < 0.01).

**Figure 6 jof-08-00974-f006:**
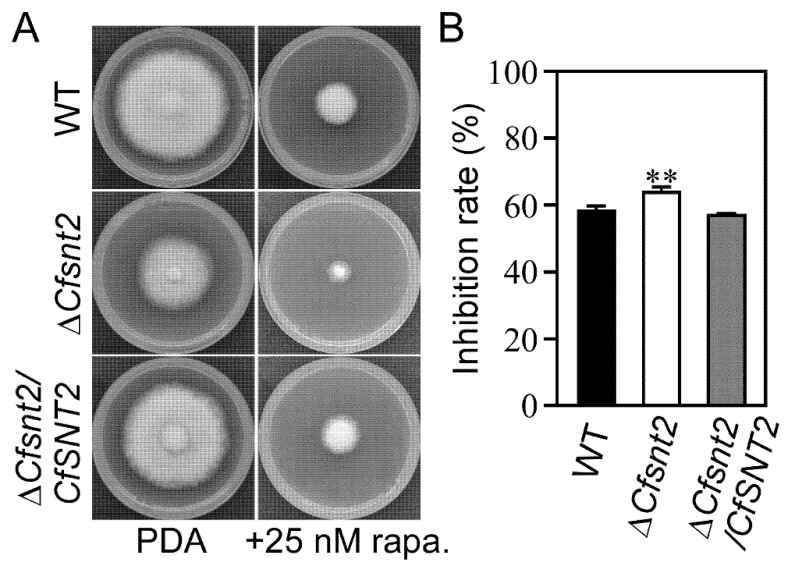
CfSnt2 is involved in the rapamycin response. (**A**) The strains of WT, Δ*Cfsnt2* mutant, and complemented strain Δ*Cfsnt2/CfSNT2* were incubated in PDA media or PDA media with 25 nM rapamycin at 28 °C for 3 days. (**B**) Statistical analysis of inhibited rates of the strains to rapamycin stress. Three independent experiments were performed, with three biological replicates each time. Error bars indicate SD of three replicates (** *p* < 0.01).

**Figure 7 jof-08-00974-f007:**
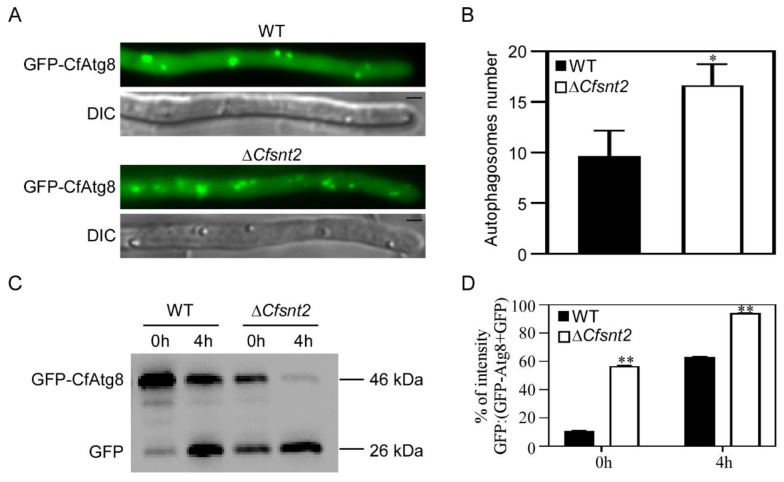
CfSnt2 negatively regulates autophagy. (**A**) Fluorescent micrographs of autophagosomes in WT and Δ*Cfsnt2* mutant. Strains expressing the *GFP-CfATG8* in WT and Δ*Cfsnt2* mutant were observed by fluorescent microscopy. Bar = 5 μm. (**B**) Statistical analysis of autophagosome numbers in hyphal tips. Error bars indicate SD and asterisk represents significant differences (* *p* < 0.05). (**C**) Immunoblot analysis of GFP-CfAtg8 proteolysis. Strains expressing the *GFP-CfATG8* in WT and Δ*Cfsnt2* mutant were cultured in MM-N for 4 h, and the mycelial proteins were extracted and immunoblotted with anti-GFP antibody. (**D**) Relative intensity of GFP/(GFP-CfAtg8 + GFP) ratios. Asterisks represent significant differences (** *p* < 0.01).

**Figure 8 jof-08-00974-f008:**
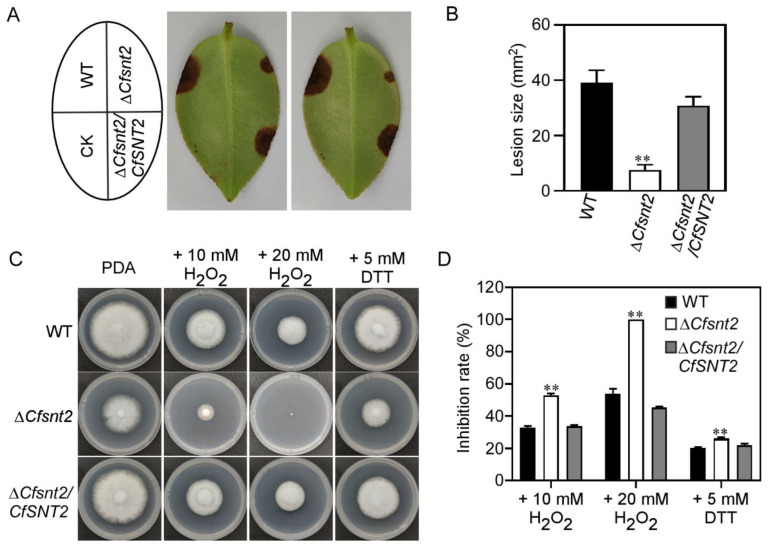
CfSnt2 regulates pathogenicity and the responses to oxidative and ER stresses. (**A**) Wounded *C. oleifera* leaves were inoculated with mycelial plugs of WT, Δ*Cfsnt2*, and Δ*Cfsnt2/CfSNT2*. Diseased symptoms were observed at 4 days post inoculation. CK: control check, agar plugs were inoculated on wounded leaves. (**B**) Statistical analysis of lesion sizes measured by Image J. Asterisks represent significant differences (** *p* < 0.01). (**C**) The strains of WT, Δ*Cfsnt2* mutant, and Δ*Cfsnt2/CfSNT2* were incubated in PDA media or PDA media with H_2_O_2_ and DTT at 28 °C for 3 days. (**D**) Statistical analysis of inhibited rates of the strains to oxidative and ER stresses. Three independent experiments were performed with three biological replicates each time. Error bars indicate SD of three replicates (** *p* < 0.01).

## Data Availability

All data supporting the findings of this study are available within the paper and its Appendix A (published online).

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
