# Peer review of "The CfSnt2-Dependent Deacetylation of Histone H3 Mediates Autophagy and Pathogenicity of Colletotrichum fructicola"

_jof, 2022, doi:10.3390/jof8090974_

Round 1

Reviewer 1 Report

The authors raise an important problem of the mechanism of pathogenicity of Colletotrichum fruticola. This pathogenic fungus causes the antheacnose of the economically important plant Camellia oleifera.

Authors showed localization of Snt2 and visualized increase in acylation of H3 - very elegant pictures

Transcriptomic analysis of the WT and snt2 strains with the deletion showed that deletion of snt2 gave a broad effect of stimulating and inhibiting the expression of many genes.

The authors assign genes to each class according to function and analyze these classes. Particular attention is paid to genes related to autophagy because almost all genes related to autophagy have been expressed upregulated.

The authors concluded that snt2 negatively regulates autophagy.

The ∆Snt2 mutant is more sensitive to oxidative stress and ER stress. There was also a statistically significant sensitivity to rapamycin compared to the wild-type strain.

The authors also demonstrated the reduced pathogenicity of the mutant strain against C. oleifera leaves

In my opinion the article is valuable and should be published

I have a few doubts which are listed below

Line 68 – an information why GFP tag fused to the C-terminus of CfSnt2 was not working should be added

Fig 6, Fig 8– what CK means

after analyzing the ATG8 expression of the autophagy marker protein, they found that the PDA culture revealed more phagosomes in the snt2 mutant. However, studies on the MM-N medium showed a reduced amount of Atg8 protein in the mutant.

I don't understand why the authors show an untaged GFP that increases over time. Where does free GFP come from in the cell.

Materials and methods

Construction of the ∆Snt2 mutant is not described

Construction of GFP-CfSnt2 and H1-RFP containing vectors for localization of the Snt2 protein is not decribed

4.1.  MM-N medium is not described in the methods section

4.3. 1 mL lysis buffer – the content ???

10 µl cocktail – protease inhibitors???

the origin of the reagents and antibodies is not described

4.6. In the Method section, the concentrations of reagents should be indicated , although I found them under the figures.

Reviewer 2 Report

This is an interesting study that showed the association between CfSnt2, autophagy and pathogenicity in Colletotrichum fructicola, linking specific hypothesised mechanisms of effect to pathogenicity related autophagy from H3 deacetylation. Overall, it is well written. I have no serious concerns, and only have some minor comments.

11- the genetic background of mutant strains must be explained (related articles are difficult to obtain because they are often only available in Chinese). For example, can you please specify if the complemented strain genome has been fully sequenced and does not contain compensatory mutations likely to explain the slight observed phenotypic differences between ∆CfSnt2/CfSNT2 and WT strain.

22- Line 86: “complemented strains”. Besides, why did you mention these two complemented strains only in figure 1? Please clarify this point.

33- Line 146: Table S1 instead of Table S2.

44- Line 174: change “GFP-CfAtg8” by ∆CfSnt2.

Reviewer 3 Report

Camellia oleifera is a valuable woody edible-oil crop in China and Colletotrichum fructicola is a dominant causal agent of anthracnose on C. oleifera. This manuscript identified a Snt2 protein and provided evidence showing that CfSnt2 negatively regulates autophagy and is involved in the responses to host-derived ROS and ER stresses. Even though lots of work is well done in this manuscript, no surprise can be found throughout this manuscript. In general, most of the investigations were superficial and descriptive. The author should explain new findings and mechanisms to expand how Snt2 regulates the autophagy pathway and pathogenicity rather than repeat previous work in other fungal pathogens. The same work was done in Magnaporthe oryzae(He et al. 2018). In addition, the author needs to add more detailed evidence to explain why the mutants reduce pathogenicity. So, I hope the author provides some novel findings in this manuscript. My detailed comments are as below:

(1) Does the CfSnt2 conservation among the eukaryotes?

(2) Transcriptomic analysis of the WT and ΔCfsnt2 mutant occupies half of this article. However, no true significance in these sections.

(3) Most of ATG genes up-regulated in ΔCfsnt2 mutant by qRT-PCR data, but the Gene ontology enrichment and KEGG pathway enrichment were not shown the autophagy pathway. How to explain it?

(4) In Fig7, ΔCfsnt2 mutant showed more autophagosomes than WT in PDA rich medium, however, the GFP-CfAtg8 band was heavy in WT rather than in ΔCfsnt2, why?

(5)What is the meaning of CK in Fig. 6A and Fig. 8A?

(6) The materials and methods are so sample.

(7) To better understand the autophagy progress in fungi, I recommend reading and citing some high level papers as below.

1.             Zhu, X.M., et al., A VASt-domain protein regulates autophagy, membrane tension, and sterol homeostasis in rice blast fungus. Autophagy, 2021. 17(10): p. 2939-2961.

2.             Zhu, X.M., et al., Current opinions on autophagy in pathogenicity of fungi. Virulence, 2019. 10(1): p. 481-489.

3.             Marroquin-Guzman, M. and R.A. Wilson, GATA-Dependent Glutaminolysis Drives Appressorium Formation in Magnaporthe oryzae by Suppressing TOR Inhibition of cAMP/PKA Signaling. PLOS Pathogens, 2015. 11(4): p. e1004851.

4.             He, M., et al., MoSnt2-dependent deacetylation of histone H3 mediates MoTor-dependent autophagy and plant infection by the rice blast fungus Magnaporthe oryzae. Autophagy, 2018. 14(9): p. 1543-1561.

Round 2

Reviewer 3 Report

The author answered my questions one by one and most of the answers were satisfactory. The new version of this manuscript has improved a lot. I think this version can be published in the Journal of Fungi after a minor revision.

Line 87, ΔCfsnt2 should use ”italic”

Line 133 pathways instead of pathway